



# Hadal water biogeochemistry over the Izu-Ogasawara Trench observed with a full-depth CTD-CMS

Shinsuke Kawagucci[1][2][3][#][*], Akiko Makabe[3][#], Taketoshi Kodama[4], Yohei Matsui[2][3], Chisato Yoshikawa[5], Etsuro Ono[6][$], Masahide Wakita[7], Takuro Nunoura[3][8], Hiroshi Uchida[6], Taichi Yokokawa[3][8][*]

[1]Department of Subsurface Geobiological Analysis and Research (D-SUGAR), Japan Agency for Marine-Earth Science and Technology (JAMSTEC), 2-15 Natsushima-cho, Yokosuka 237-0061, Japan.
[2]Research and Development Center for Submarine Resources (SRRP), Japan Agency for Marine-Earth Science and Technology (JAMSTEC), 2-15 Natsushima-cho, Yokosuka 237-0061, Japan.
[3]Project Team for Development of New-Generation Research Protocol for submarine resources, Japan Agency for Marine-Earth Science and Technology (JAMSTEC), 2-15 Natsushima-cho, Yokosuka 237-0061, Japan.
[4]Japan Sea National Fisheries Research Institute, Japan Fisheries Research and Education Agency, Niigata, Japan, 5939-22 Suido-cho, Niigata 951-8121, Japan.
[5]Department of Biogeochemistry, Japan Agency for Marine-Earth Science and Technology (JAMSTEC), 2-15 Natsushima-cho, Yokosuka 237-0061, Japan.
[6]Research and Development Center for Global Change (RCGC), Japan Agency for Marine-Earth Science and Technology (JAMSTEC), 2-15 Natsushima-cho, Yokosuka 237-0061, Japan.
[7]Mutsu Institute for Oceanography, Japan Agency for Marine-Earth Science and Technology (JAMSTEC), 690 Kitasekine, Mutsu 035-0022, Japan
[8]Research and Development Center for Marine Biosciences, Japan Agency for Marine-Earth Science and Technology (JAMSTEC), 2-15 Natsushima-cho, Yokosuka 237-0061, Japan.
[$]Present address: Japan Meteorological Agency, 1-3-4 Otemachi, Tokyo 100-8122, Japan.
[#]Equally contributed.

*Correspondence to*: Shinsuke Kawagucci (kawagucci@jamstec.go.jp)
     & Taichi Yokokawa (taichi.yokokawa@jamstec.go.jp)

**Abstract. (174 words)** Full-depth profiles of hydrographic and geochemical properties at the Izu-Ogasawara Trench were observed for the first time using a CTD-CMS (Conductivity Temperature Depth profiler with Carousel Multiple Sampling) system, supplemented by some comparative samplings at the northern Mariana Trench using the same methods. The CTD sensor, calibrated from measurements of seawater samples, demonstrated a well-mixed hydrographic structure below 7,000 m within the Izu-Ogasawara Trench. Seawater samples collected from this well-mixed hadal layer exhibited constant concentrations of nitrate, phosphate, silicate, and nitrous oxide as well as constant nitrogen and oxygen isotopic compositions of nitrate and nitrous oxide. These results agree well with previous observations of the Izu-Ogasawara hadal waters and deep-sea water surrounding the Izu-Ogasawara Trench. In turn, methane concentrations and isotopic compositions indicated spatial heterogeneity within the well-mixed hadal water mass, strongly suggesting a local methane



source within the trench, in addition to the background methane originating from the general deep-sea bottom water. Sedimentary compound releases, associated with sediment re-suspensions, are considered to be the most likely mechanism for generating this significant $CH_4$ anomaly.

## 5    1 Introduction

The CTD-CMS (Conductivity Temperature Depth profiler with Carousel Multiple Sampling) system has been the most essential and fruitful device in the history of oceanography. Seawater sensing by the CTD draws a continuous profile of the general hydrographic property, while seawater sampling at multiple selected depths by the CMS allows a variety of subsequent analyses for chemical and biological properties. Cross-ocean observations with the CTD-CMS, in some case led

by international programs such as WOCE (Siedler et al., 2001) and GEOTRACES (Cutter, 2013), have yielded datasets of comparable qualities and sufficient horizontal resolution that serve as a basis for us to paint the overall picture of the Earth's ocean. However, one blank gap remains in that picture – the hadal waters (deeper than 7,000 m), due to the extreme pressure there and a lack of full-depth CTD-CMS systems.

The lack of full-depth CTD-CMS systems greatly limited seawater sampling-based analyses and therefore the analysis and understanding of hadal water column in a way comparable to the CTD-CMS data available for all other parts of the ocean. Nevertheless, a small number of hadal sampling efforts, not using the CTD-CMS, have been carried out over the years. For example, a pioneering work (Mantyla and Reid, 1978) collected a total of four hadal seawater samples from two stations in the southern Mariana Trench (MT) with a specially-designed acoustic release device. Hydrographical properties (salinity,

oxygen, and nutrients) of the samples obtained demonstrated that hadal and abyssal waters exhibit identical characteristics.

At a trench axis station AN1 (29°05N - 142°51E, 9,750 m) located in the mid Izu-Ogasawara Trench (IOT), vertical seawater samples were collected during two cruises of R/V *Hakuho-maru* in 1984 and 1994 (Nozaki et al., 1998; Gamo and Shitashima, 2018), by directly attaching standard Niskin bottles to a wire line. The samples were used to determine

hydrographic properties and for [222]Rn analyses, while three specially-designed samplers were also used for metals and other radionuclides analyses. The hadal water at AN1 exhibited vertically constant profiles in salinity, dissolved oxygen, nitrate, and phosphate, confirming the characteristics previously observed in the southern MT (Mantyla and Reid, 1978). The renewal time of the trench-filling hadal water was evaluated to be ~5 years from the distribution of radionuclides (Nozaki et al., 1998). Manganese and iron were slightly enriched in the hadal water only and was background level in the overlying

abyssal water while significant [222]Rn excess was detected in waters up to ~2,700 m above the trench axis seafloor, suggesting lateral [222]Rn supply from the surrounding trench slope (Gamo and Shitashima, 2018). These hadal water-specific characteristics were consistent with a scenario where the transport of sedimentary component from the trench slope to hadal



depths being frequent, which has indeed been commonly shown by sediment observations at trench axes of Izu-Ogasawara, Mariana, Japan, and Tonga trenches (Nozaki and Ohta, 1993; Glud et al., 2013; Oguri et al., 2013; Wenzhofer et al., 2016) and supported by numerical modeling (Ichino et al., 2015).

In addition to the geochemical studies outlined above, ~30 vertical seawater samples, including 7 hadal samples, were collected mainly for biological analyses by the remotely operated vehicle *ABISMO* at the Challenger Deep in the southern MT. Phylogenetic analyses from these samples revealed a hadal water-specific microbial community, named the hadal biosphere, which has been shown to be distinct from the abyssal one (Nunoura et al., 2015). Mechanisms developing the hadal biosphere are considered to be likely associated with the lateral supply of sedimentary organic matter from the trench

slope as well as the hadal geochemical anomalies.

Despite these efforts, the spatial coverage within and among trenches, as well as the comprehensiveness of the (bio)geochemical dataset, is still greatly limited compared to other parts of the ocean. To resolve these problems and to obtain data from hadal depths in a directly comparable quality to the global CTD-CMS observations, the development of a

full-depth CTD-CMS system is needed and indeed has long been awaited. At last, a full-depth-rated CTD-CMS system was equipped in 2015 on the research vessel R/V *Kaimei* launched that year. The R/V *Kaimei* CTD-CMS system has been deployed at the IOT and the MT including 5 full-depth vertical sequences at trench axes. Here we report the results of (bio)geochemical analyses for these CTD-CMS collected seawater samples, including a total of 57 samples from hadal depths, mostly in the IOT.

## 2. Settings and Methods

### 2.1   Study area and equipment

Seawater column of the IOT and the MT were investigated during 3 R/V *Kaimei* cruises (KM16-02, KM16-08, and KM17-01) and a R/V *Shinsei-maru* cruise (KS-16-8). Locations of the sampling stations are shown in Table 1 and Figure

1. Although the term "hadal" generally refers to the zone below 6,500 m (Jamieson 2015), in this study we use "hadal" for depths below 7000 m and "abyssal" for depths between 4,000-7,000 m, based on the vertical hydrographic constitution revealed by our observations (see Section 3.1). Seawater sampling was generally conducted at 50-250 m vertical intervals at each station.

R/V *Kaimei* has a 12,000m-long synthetic-fiber coaxial cable for the operation of its full-depth CTD-CMS system, which consists of a CTD sensor (SBE911plus, 11,000 m capable), a Carousel water sampler (SBE32, 11,000 m capable) for 36 Niskin-X bottles, and a dissolved oxygen sensor (RINKO, 7,000 m guaranteed but 10,000 m capable). The CTD



thermometer (SBE3plus) was calibrated in situ with a deep ocean standard thermometer (SBE35, 7,000 m capable) to correct a pressure dependency of the CTD thermometer (Uchida et al., 2015). The full dataset of the CTD observations will be deposited on a website and discussed elsewhere (Uchida et al., in prep.). R/V *Shinsei-maru* has an 8,000m-long steel coaxial cable for the operation of its CTD-CMS system (6,500 m capable), consisting of a CTD sensor (SBE911plus), a Carousel

water sampler (SBE32), 24 Niskin-X bottles, and a dissolved oxygen sensor (SBE43).

## 2. 2   Analyses

The potential density anomaly ($\sigma_{6K}$: pref = 6,000 dbar, potential density minus 1,000 kg/m$^3$) was calculated from the CTD dataset using the TEOS-10 software, calibrated using practical salinity measured. Dissolved oxygen was also measured on the ship using a modified Winkler's titration method with a precision within 0.2 µmol/kg, estimated from standard deviation

of replicate measurements.

Nitrate+nitrite (hereafter, nitrate), phosphate, and silicate concentrations were determined using AutoAnalyzerII (for nitrate, Technicon, Dublin, Ireland) and TrAACS 800 (for silicate and phosphate, Bran+Luebbe, Norderstedt, Germany) in Japan Sea National Fisheries Research Institute (Hydes et al., 2010; Kodama et al., 2015). Nutrient samples were frozen in a deep

freezer at -80°C immediately after subsampling on-board and stored below -20°C until measurements were taken. Another set of samples was collected during the KM16-08 cruise just for silicate analysis, which were stored at 4°C in the dark and analyzed within two weeks of sampling. In-house standards, prepared using artificial seawater, were used in combination with commercially available nutrient standard solutions (RMNS, KANSO Technos, Tsukuba, Japan) (Aoyama et al., 2012), but only RMNS were used for the standardization of concentrations. RMNS lots of BD, BZ, and BS were used for samples

taken during the cruise KM16-02, while BT, BS, and CB were used for KM16-08, and BY, CA, and CB for KM17-01. Phosphate and silicate data could not be obtained for samples taken during KM17-01 due issues with the instrument. The determination limits of nutrient concentrations, defined as tripled standard deviations of blank values, were <0.1, <0.05, and <1 µmol/kg for nitrate, phosphate and silicate, respectively. Analytical errors in our measurement systems, estimated by repeated analyses (*n* = 5) using the BV lot of RMNS, were 0.48, 0.04, and 0.21 µmol/kg for nitrate, phosphate, and silicate,

respectively, as previously reported in the inter-laboratory comparisons of seawater nutrient concentrations (see Aoyama et al., 2016). Seawater for ammonium analysis was subsampled into a 25 mL polypropylene bottle after triple-rinsed with seawater, and ammonium concentration was measured onboard using a fluorometric (OPA) method (Holmes et al., 1999). The detection limit of ammonium concentrations was as low as 10 nM.

Nitrogen and oxygen isotopic compositions of nitrate were analyzed by the denitrification method (Casciotti et al., 2002; McIlvin and Casciotti 2011) at Japan Agency for Marine-Earth Science and Technology (JAMSTEC). A subsample (~50 mL) of seawater was filtered on-board with a 0.45 µm cellulose acetate membrane filer (ADVANTEC) and stored at -20°C. For surface seawater (<500 m) in which nitrite was comparable to nitrate, sulfamic acid was injected to sample seawater to



eliminate nitrite as NO volatile (Granger and Sigman, 2009). The seawater samples were then injected into a laboratory denitrification culture in 20 mL headspace glass vials. Nitrate (and nitrite) was quantitatively converted to nitrous oxide using the strain *Pseudomonas chlororaphis* (JCM20509 = ATCC13985), a denitrifying bacterium lacking the ability to reduce $N_2O$. International reference materials (IAEA-N3, USGS-32, and USGS-34 for both $\delta^{15}N_{NO3}$ and $\delta^{18}O_{NO3}$, and USGS-

35 for $\delta^{18}O_{NO3}$; Böhlke et al., 2003) and in-house working standard were included within a batch of 120 bottles for a series of analysis. The $N_2O$ produced from each culture was then analyzed using a ThermoFinnigan GasBench+PreCon trace gas concentration system, interfaced to a Delta V Plus isotope-ratio mass spectrometer. The conversion efficiency from nitrate to $N_2O$, evaluated by comparing with the TrAACS 800 analysis, was above 95% in this study. Analytical precisions were estimated to be within 0.2 ‰ for $\delta^{15}N_{NO3}$ and 0.3 ‰ for $\delta^{18}O_{NO3}$.

The potential nitrification activity was estimated during the KM16-08, KM17-01, and KS-16-8 cruises by ship-based incubation of seawater with 3 [15]N-labeled substrates (ammonium, urea, and glutamine) according to an established method (e.g., Shiozaki et al., 2016). Deep-sea water samples were subsampled into 125 mL pre-combusted glass vials. Each of [15]N-ammonium, -urea, and -glutamine (amide group-labeled) were then spiked into the vials to give a final concentration of 50

nM, and the samples were incubated at 4 $^{o}$C in the dark for 5-10 days prior to filtration (0.45 μm cellulose acetate membrane filer; ADVANTEC). The nitrogen isotopic ratio of the nitrate+nitrite within the filtrate was measured according to same method as for native nitrate analyses, as described above. For comparative purposes, seawater from shallower than 1,000 m were also examined by a similar method, with incubation being carried out under *in situ* temperatures and for a short duration of ≤12 hours. For the calculation of nitrification activity, it was assumed that the concentrations of ammonium, urea,

and glutamine were negligible in the native deep-sea water. The detection limit of nitrification activity, determined based on significant changes in the nitrate isotope ratios (>0.6‰, three times higher than the analytical uncertainty) between [15]N-spiked and non-spiked seawaters after the incubation, was estimated to be 0.02 nmol/L/day.

The total organic carbon (TOC) concentration was determined using a total organic carbon analyzer (TOC-L, Shimadzu Co.,

Kyoto, Japan) following a published analytical method (Wakita et al., 2016), with some minor modifications. To determine the TOC, instead of dissolved organic carbon (Wakita et al., 2016), seawater samples were not filtrated on the ship. The TOC concentrations were calibrated against consensus reference materials provided by Prof. D. A. Hansell (Miami University) and determined to a precision of ±1.0 μM. The radiocarbon content ($\Delta^{14}C$) of the dissolved inorganic carbon (DIC) was measured, following the methods published by Kumamoto et al. (2011), at the Institute of Accelerator Analysis

Ltd in Shirakawa, Japan (using a Pelletron 9SDH-2, National Electrostatics Corporation, USA) with a 4‰ analytical error.

Concentrations and isotopic compositions of nitrous oxide ($N_2O$) and methane ($CH_4$) were determined according to published methods (Hirota et al., 2010) with some modifications for $CH_4$ purification (Okumura et al., 2016a). Each seawater sample was introduced into a 120 mL glass vial from the Niskin bottle, sealed with a butyl rubber septum and



aluminum cap after the addition of 0.2 mL mercury chloride-saturated solution for poisoning, and stored at 4 $^{\circ}$C until the measurement. $N_2O$ and $CH_4$ dissolved in the seawater were extracted and purified by purge-and-trap gas chromatography prior to the introduction into a MAT253 isotope-ratio mass spectrometer. Resistors of MAT253 Faraday cups for monitoring m/z of 44, 45, and 46 for $CO_2$ derived from $CH_4$ oxidation were set respectively at $3\times10^9$, $3\times10^{11}$, and $1\times10^{12}$ Ohms, instead

of the generally used $3\times10^8$, $3\times10^{10}$, and $1\times10^{11}$ Ohms. This modification improved the detection limit of $CH_4$ to as low as 0.08 nmol/L, while keeping the comparable analytical precision. Errors for analyses conducted during the present study were estimated from repeated analyses of a sample, and were 10% for $N_2O$ concentration, 0.2‰ for $\delta^{15}N_{N2O}$, 0.5‰ for $\delta^{18}O_{N2O}$, 20% for $CH_4$ concentration, and 0.3‰ for $\delta^{13}C_{CH4}$.

To examine the validity of our sampling and analyses, nutrients concentrations of abyssal waters at neighbor stations obtained by the WOCE re-occupation program (Stn. 65-90 on P10 line and Stn. 273-321 on P03-revisit line) are selected (Kawano and Uchida, 2007a, 2007b). Basin-scale $\Delta^{14}$C-DO relation below 4,000 dbar from stations of the WOCE P01 (Kawano et al., 2009) and P03 (Kawano and Uchida, 2007a) in addition to P02 (https://cchdo.ucsd.edu/cruise/318M200406) is also used for interpretation.

All analytical results, full-depth vertical profiles, and the overall summary of hadal water analyses are presented in Supplementary Table S1, Supplementary Figure S1, and Table 2, respectively. As the present study focuses on observations and data from deep waters, properties of the surface water are generally used for validating the observations and only discussed minimally.

## 3. Results

### 3.1 Major hydrographic property

Vertical profiles of the potential density anomaly ($\sigma_{6K}$) revealed that a dense, well-mixed water mass occupies the deeper parts of the trenches (Figure 2a), which we call the "hadal mixed layer". The boundary between this layer and the abyssal

layer above is seen as a change in $\sigma_{6K}$ three times greater than a standard deviation within the nearly-constant density of the hadal mixed layer. This boundary was shallower than 7,000 m in all axis stations (Figure 2a), and we therefore redefine hadal waters as being deeper than 7,000 m in this study.

Both potential density and DO of the hadal water at the MT station (54.356 and 172 μmol/kg) were higher than those at the

IOT stations (54.350 and 164 μmol/kg) (Figure 2a and 2b). These clear differences in properties between hadal waters in the two trenches are in line with the current knowledge from abyssal water observation as northward bottom current in the western Pacific (Wijffels et al., 1998) and the presence of a geographical barrier between IOT and MT (Ogasawara Plateau,



Figure 1). Detailed analyses of the CTDO dataset, in addition to its comparison with past CTD(O) observations (Taira et al., 2005; Taira, 2006; Gamo and Shitashima, 2018), will be presented elsewhere (Uchida et al., in preparation).

### 3.2 Nitrogen biogeochemistry (nitrate, $N_2O$, ammonium, and potential nitrification activity)

Vertical profiles of nitrate concentrations agreed well among all stations (Figure 2c). Nitrate concentrations of hadal and abyssal waters at IOT stations were almost constant at 35.20±0.41 µmol/kg (n = 47) and 35.50±0.68 µmol/kg (n = 149), respectively (Table 2). The same for the MT stations were 34.72±0.05 µmol/kg (n = 10) and 35.12±0.38 µmol/kg (n = 16), respectively. The nitrate variations in the hadal waters are smaller than the analytical error (0.48 µmol/kg) while a greater nitrate variation in the IOT abyssal water likely reflects a vertical gradient within abyssal zone. Nitrate concentrations of the abyssal waters presented herein were found to be consistent with those at neighboring stations in WOCE P03 and P10 lines (35.36±0.48 µmol/kg; n = 315), confirming accurate quantification. Furthermore, the hadal water value for the IOT agrees well with those observed at AN1 in 1984 (35.1±0.3 µmol/kg) (Gamo and Shitashima, 2018).

Nitrogen and oxygen isotopic compositions of nitrate in the IOT (Figure 2d, 2e) were also found to be constant through hadal and abyssal layers at +5.10±0.15‰ and +2.22±0.11‰ (n = 25, hadal) and +5.10±0.14‰ and +2.32±0.20‰ (n = 28, abyssal), respectively (Figure 2 and Table 2). Isotopic characteristics of the deep seawater at the IOT region was consistent with the hadal water at the north MT (as reported herein), the Challenger Deep (Nunoura et al., 2015), and other published reports of deep seawater from the Pacific (e.g., Casciotti, 2016). For waters obtained above 500 m where nitrate concentrations were low (<20 µmol/kg), the isotopic compositions varied between +3‰ and +11‰ ($\delta^{15}N_{NO3}$) and +2.5‰ and +22‰ ($\delta^{18}O_{NO3}$), respectively.

Identical vertical profiles were seen for both $N_2O$ concentrations and isotopic compositions, throughout all stations (Figure 2f, 2g, 2h). $N_2O$ concentrations, $\delta^{15}N_{N2O}$, and $\delta^{18}O_{N2O}$ in hadal and abyssal waters of the IOT were constant at 17.7±1.3 nM, +8.8±0.2‰, and +51.6±0.3‰ (n = 47, hadal) and 18.1±1.0 nM, +8.9±0.2‰, +51.8±0.4‰ (n = 99, abyssal), respectively (Figure 2 and Table 2). These characteristics of deep-sea $N_2O$ in the IOT region are consistent with those in the MT (this study) and other stations in the north Pacific such as ALOHA (Popp et al., 2002) and KNOT (Toyoda et al., 2002). In turn, variable concentrations (5-42 nM) and isotopic composition (+6.2-+9.5‰ and +44-+57‰ for $\delta^{15}N_{N2O}$ and $\delta^{18}O_{N2O}$, respectively) were seen in $N_2O$ above 4,000 m depth in the IOT region, which were not identical with other deep seawater in the Pacific (Toyoda et al., 2017).

Ammonium concentrations at the IOT were not detectable (<10 nM) for both hadal and abyssal waters, while at the surface it was as high as 30 nM. It is consistent with hadal waters from the Challenger Deep that also had negligible ammonium concentration (Nunoura et al., 2015).



The potential nitrification activity of the hadal and abyssal waters in all samples analyzed for the present study was below the determination limit ($\delta^{15}N_{NO3}$ increase smaller than 0.6‰). Even if the $\delta^{15}N_{NO3}$ increase observed (<0.6‰) was significant, the nitrification activity should be not higher than 0.02 nmol/L/day (Figure 3 and Supplementary Table S2). The nitrification

activity of the surface water was up to 5 nmol/L/day and comparable to those reported from other oligotrophic stations such as ALOHA and BATS (1-8 nmol/L/day) (Beman et al., 2011), which serves to confirm the validity of our data. At DO minimum depth (~1,500 m), several incubations exhibited detectable nitrification activity (Figure 3).

### 3. 3    Phosphate, silicate, radiocarbon, and TOC

Phosphate concentrations at each station were generally constant through hadal and abyssal depths (Figure 2i). For example, samples from the KM16-08 cruise exhibited phosphate concentrations of 2.40±0.01 µmol/kg for hadal waters (n = 11) and 2.42±0.02 µmol/kg for abyssal waters (n = 21), respectively, demonstrating a lack of hadal-specific phosphate cycle. Hadal phosphate concentrations during the KM16-08 cruise were consistent with those reported for hadal waters at AN1 in 1984 (2.39±0.04 µmol/kg)(Gamo and Shitashima, 2018) and for abyssal water (below 5,500 dbar) at neighboring stations of

WOCE P03 and P10 (2.44±0.02 µmol/kg, n = 77). A few systematic differences seemed present in phosphate concentrations among the different cruises (Figure 2i) while the N-P relationships within each cruise were clearly linear (Supplementary Table S1).

Silicate concentrations were highly variable among the stations (Figure 2j), probably due to freezing and thawing of the

samples, as discussed previously by Zhang and Ortner (1998). Compared to silicate distributions of the WOCE P03 and P10 lines, samples maintained at 4°C from the KM16-08 cruise showed consistent vertical profiles but the -20°C frozen samples from the other cruises generally showed lower concentrations. At stations sampled during KM16-08 (34N02 and 34N03), silicate concentrations were constant within the hadal water (145.6±0.2 µmol/kg, n = 11) but slightly variable in the abyssal water (148.9±2.3 µmol/kg, n = 21) due to the presence of a vertical gradient. Concentrations measured from samples

obtained at KM16-08 stations were consistent with those reported from LM2 (~143 µmol/kg) (Gamo and Shitashima, 2018) but slightly higher than those of the abyssal water (below 5,500 dbar) at neighboring stations of WOCE P03 and P10 (134.7±2.43 µmol/kg, n = 77).

The radiocarbon contents ($\Delta^{14}C$) of the hadal and abyssal waters at the IOT and the MT were constant within each water

layer: -206.6±4.4‰ (n = 40, IOT hadal), -209.8±7.4‰ (n = 62, IOT abyssal), -197.0±3.9‰ (n = 10, MT hadal), and -210.0±5.8‰ (n = 16, MT abyssal) (Figure 2k and Table 2). The variation in $\Delta^{14}C$ for hadal depths, comparable to the analytical error (±4‰), is reasonable because the change in $\Delta^{14}C$ is expected to be <1‰ from the estimated renewal time of ~5 yr (Nozaki et al., 1998) and the $\Delta^{14}C$-yr relationship of 8.7 yr/‰ (Sarmiento and Gruber, 2006). The DO-$\Delta^{14}C$





relationships within the hadal water mass ranged within those of the Pacific deep seawater from WOCE P01-P03 (Figure 4). Variations of DO and $\Delta^{14}$C within the hadal water mass were too small for evaluating trench-specific oxygen consumption rates, but the DO-$\Delta^{14}$C relationship of the Pacific deep seawater yields a slope of 0.749±0.030 μmol/kg/‰, corresponding to an oxygen consumption rate of 0.0860±0.0034 μmol/kg/yr.

The concentrations of TOC in the deep seawater (below 2,000 m) of the IOT and MT regions were rather constant at ~38 μM with the exception of a few samples (Figure 2l and Table S2). This TOC level agrees well with DOC observations at a neighboring abyssal plain station (S1: 30°N - 145°E) (http://ebcrpa.jamstec.go.jp/k2s1/en/index.html). The TOC concentrations at the stations 34N02 and 31N03 appear to be increasing from 8,000 and 9,000 m towards the bottom,

respectively, despite the well-mixed hydrographic property. The observed increases are significant beyond the analytical uncertainty of ±1 μM (Figure 2l). The hadal TOC gradient strongly suggests inner-trench TOC input since negligible TOC catabolism is expected from the negligible DO change (<0.1 μmol-$O_2$/kg). Abyssal samples collected at 32N during the KS-16-8 cruise also exhibited TOC concentrations higher than 40 μM.

**3. 4  Methane**

Methane concentrations were generally as low as 0.25 nM below 2,000 m deep (Figure 2m). $CH_4$ concentrations lower than 1 nM are ubiquitous in deep-sea water (Hirota et al., 2010; Son et al., 2014) and is probably regulated by a minimum threshold in microbial uptake for aerobic methanotrophy. Opposite to the general trend, an increase of methane concentrations to three times higher than the general level was detected in waters below 8,000 m deep at the station 34N02

(Figure 2m), despite well-mixed hydrographic property.

$\delta^{13}$C values of $CH_4$ were highly variable between -60‰ and -30‰ (Figure 2n), despite there being little variation in concentrations among the sites (Figure 2m). A clear spatial trend in $\delta^{13}C_{CH4}$ values was observed in the IOT region (Figure 5). Vertically, $\delta^{13}C_{CH4}$ values were ~-38‰ between 4,000-6,000 m and then generally decreased with increasing depth, and

became constant below 8,000 m (Figure 2n). Below 8,000m, $\delta^{13}C_{CH4}$ values exhibited a north-south trend where the values were lower (-56.2±1.4‰: n = 7) at 34N than the other IOT stations (-46.3±2.2‰: n=23) and also the MT stations (-35.5±3.7‰: n = 5)(Figures 2n and 5). Moreover, values measured from abyssal waters suggested a west-east trend, with $\delta^{13}C_{CH4}$ from the western stations (relative to the axis: Figure 1) being generally lower than -40‰ and from the eastern stations generally higher than -40‰ (Figure 5).




## 4. Discussion

### 4.1 Sedimentary compound release through sediment re-suspension at slopes

The $CH_4$ heterogeneity, despite the well-mixed hydrographic property, in the IOT hadal water strongly suggests the occurrence of local $CH_4$ source(s) within the trench, in addition to the background $CH_4$. The background $CH_4$ has a
concentration of ~0.25 nM with $\delta^{13}C_{CH4}$ of -38‰, while the $CH_4$ from the local source would have $\delta^{13}C_{CH4}$ of ~-58‰, as observed in the 34N hadal water. The $^{13}C$-moderate $CH_4$ observed in the hadal waters from 27-32N can be attributed to the kinetic isotope effect of microbial methane consumption, which causes $^{13}C$ enrichment in the remnant $CH_4$ (Tsunogai et al., 2000; Feisthhuer et al., 2011), in addition to water mass mixing (Figure 6). Assuming the endmember composition of the local $CH_4$ source within the trench being 1.0 nM with $\delta^{13}C_{CH4}$ of -58‰ and taking the probable range of kinetic isotope
effects ($\varepsilon$ = 5‰ - 20‰) (Tsunogai et al., 2000; Feisthhuer et al., 2011) into consideration, the compositional and isotopic variation of $CH_4$ observed in the IOT can be reasonably explained (Figure 6). The microbial consumption of $CH_4$ in sub-nano-molar quantities, if it occurs, is not in conflict with the negligible change in DO observed, the detection limit of which is ~100 nmol/kg. Multiple $CH_4$ local sources with different $^{13}C$ signatures, other than the 34N source proposed above, are also a possibility in generating the $\delta^{13}C_{CH4}$ variations observed within the IOT hadal water, and does not conflict with the
discussion above.

The observed $^{13}C$-depleted $CH_4$ at the northern and western stations appears to be consistent with known two bottom water currents. In addition to a generally northward-flowing basin-scale bottom current, a local counter-clockwise current at ~6,000 m deep in the IOT was revealed by previous current meters deployment on the seafloor at 34N (Fujio et al., 2000).
The basin-scale current brings the background $CH_4$ into the eastern sill of the IOT. On the other hand, the local counter-clockwise current, particularly southward current along with western slope, distributes the $^{13}C$-depleted $CH_4$-rich water occurring at 34N to the western part of inner trench water, resulting in $\delta^{13}C_{CH4}$ values being lower on the western side on the 31N line. Also, the north-south trend in $\delta^{13}C_{CH4}$ below 8,000 m mentioned above can be explained by a north-south mixing between the local $^{13}C$-depleted $CH_4$ at 34N and the background $CH_4$ from the Ogasawara Plateau region. Nevertheless,
multiple $CH_4$ sources in addition to the proposed 34N source can also generate the observed spatial heterogeneity in $\delta^{13}C_{CH4}$ values.

Several local $CH_4$ sources are known in the deep-sea, but it appears very unlikely for most of them to be accountable for the hadal $CH_4$ and other geochemical anomalies seen in the IOT, such as TOC, Mn, and $^{222}Rn$ (Gamo and Shitashima, 2018).
Firstly, the discharge of high-temperature hydrothermal fluid is a well-known source for deep-sea $CH_4$ plumes that are accompanied with Fe, Mn, and $^{222}Rn$ (e.g., Kadko et al. 1990; Resing et al. 2015). However, hydrothermal vents in the Izu-Ogasawara region are only expected to occur at the volcanic arc and back arc with depths of ~1,500 m (e.g. Tsunogai et al. 1994). Secondly, cold fluid seepage at accretionary prisms contains abundant $CH_4$ (e.g., Toki et al. 2004), but is unlikely to





occur at the IOT which is non-accretionary. Thirdly, serpentinization-associated $CH_4$-rich geofluids have been reported from seafloors where (sea)water penetrates into peridotite (Fryer 2011; Kelley et al. 2005). An example has been reported from the terrestrial-ward slope of the Challenger Deep, southern MT (Okumura et al., 2016b). The exposure of serpentinized peridotite below 5,500 m at the terrestrial-ward slope of the IOT (Morishita et al., 2011; Harigane et al., 2013) indicates that

discharge of $CH_4$-rich geofluids occurring in the IOT cannot be ruled out, but such geofluid systems have never been discovered before. Seafloor serpentinization geofluids reported so far exhibit $\delta^{13}C_{CH4}$ values higher than -40‰ (Konn et al., 2015) and are not compatible with the hadal $CH_4$ anomaly observed in this study. Also, it appears unlikely for highly alkaline serpentinization-based geofluids to generate Mn and Fe plumes. In addition to the fluid releases, *in situ* microbial methanogenesis in deep-sea water column, which could generate $^{13}$C-depleted $CH_4$ if it does occur (e.g., Okumura et al.,

2016a), is also very unlikely due to high DO levels through the water column in the IOT (e.g., Reeburgh, 2007).

Sedimentary compounds released into the water column, through sediment re-suspension, is therefore the most likely process responsible for generating the $CH_4$ (and TOC) anomaly in the IOT water mass, as previously proposed for other hadal-specific geochemical anomalies in Mn and $^{222}$Rn (Gamo and Shitashima, 2018) and microbiological characteristics (Nunoura

et al., 2015). Earthquake-induced sediment re-suspension and the associated increase in turbidity as well as anomalies in $CH_4$ and Mn have been occasionally reported in the deep-sea water column (Tsunogai et al., 1996; Gamo et al., 2007; Kawagucci et al., 2012). Values of $\delta^{13}C_{CH4}$ ranging from -80‰ to -60‰ were observed in the turbid seawater (Kawagucci et al., 2012) and the shallow part of seafloor sediment (Ijiri et al., 2008; Heuer et al., 2009), compatible to that of the hadal water at the 34N stations. The increase in TOC observed is attributable to both suspended organic particles directly entrained into the

Niskin bottles and the dissolved organic matter stripped from the suspended particles (Komada and Reimers, 2001).

Explanation for a vertical layer ~2,000 m in thickness exhibiting anomalously high $CH_4$ (this study) and $^{222}$Rn (Gamo and Shitashima, 2018) requires sedimentary compounds to be released not only from the axis bottom, but also from the trench slopes. Re-suspensions from the trench slope have been commonly suggested by sediment observations that identified rapid

accumulation of the axis sediment, in association with frequent sediment transports from slopes (Nozaki and Ohta, 1993; Glud et al., 2013; Oguri et al., 2013). As earthquake-induced turbid seawater was indeed evident over the trench slope after earthquake events (Gamo et al., 2007; Kawagucci et al., 2012), this geological force is probably an episodic driver for re-suspension in the IOT. On the other hand, oceanographic forces, such as tidal oscillations and eddy diffusions associated with bottom currents, are weaker but more ubiquitously occurring drivers of re-suspension in the overall deep-sea

environment including the hadal trench. Intense nepheloid bottom layers over the abyssal plain, also generated by sediment re-suspension, have been found in regions where bottom current regimes are both strong and variable (e.g. de Madron et al., 2017). The strong bottom current at the IOT sills (Fujio et al., 2000) combined with complex seafloor topography of the trench walls (Figure 1), therefore, likely facilitates re-suspension in the IOT. Nevertheless, the spatio-temporal distribution



of the sediment re-suspension events within a trench, magnitudes and durations of the relevant geochemical anomalies in hadal and abyssal waters, and mechanisms driving such re-suspensions, remain poorly understood overall.

## 4.2 Energetics of the hadal biosphere

The microbial community inhabiting hadal waters has been shown to be distinct from the abyssal one (Nunoura et al., 2015). For the formation of the hadal biosphere, a certain amount of energy is necessarily required. The minimum energy required to shift one microbial community to another without cell density change should be not lower than the energy to maintain cell density of the microbial community. The cell-specific energy requirement for 'maintenance', when cells only carry out basic metabolic functions and activities without new growth, has been suggested to be $10^{-18}$ kJ/cell/sec (= $3\times10^{-11}$ kJ/cell/yr)

(Hoehler, 2004). On the other hand, the potential energy yield of the IOT hadal-abyssal water through ammonium oxidation can be calculated from the maximum rate estimated in the present study (<0.02 nmol-N/L/day (= <7.3 nmol-N/L/yr)) and the Gibbs free energy of aerobic oxidation of ammonium to nitrate (305 kJ/mol-N), to be $<2.2\times10^{-6}$ kJ/L/yr. Evaluation of the energy potentials for maintenance ($3\times10^{-11}$ kJ/cell/yr) and yield ($<2.2\times10^{-6}$ kJ/L/yr) indicates that the ammonium oxidation in the hadal-abyssal water column can maintain $<10^5$ cell/L. Since cell densities in the hadal-abyssal water column are

typically in the order of $10^6$ cell/L (e.g., Nunoura et al., 2015), this energetic calculation suggests that, in terms of energy flux, the ammonium oxidation would only occupy a minor role in characterizing the hadal ecosystem. In turn, the oxygen consumption rate in deep Pacific seawater ($0.086\pm0.003$ µmol-$O_2$/kg/yr), evaluated from the $\Delta^{14}$C-DO relationship (see Section 3.4), is approximately an order of magnitude higher than the maximum rate possible for the ammonium oxidation-specific oxygen consumption (<0.018 µmol-$O_2$/kg/yr). If the DO-normalized potential energy yields are the same between

aerobic catabolism of deep-sea TOC and ammonium oxidation, the TOC catabolism yields $1.4\times10^{-5}$ kJ/kg/yr. This potential energy yield provides a reasonable explanation for the density of microbes observed in the deep-sea water column (~$10^6$ cell/L). Although this yield may be insufficient for explaining the rapid turnover of microbial community from the abyssal water community into the hadal biosphere, we are aware that the accurate evaluation of the actual yield requires some more factors to be determined for the hadal water. For examples, the *in situ* oxygen consumption rate in the IOT hadal-abyssal

water (assumed to be the same as Pacific deep-sea-averaged level in the above calculation) and the actual potential energy yield of TOC catabolism (assumed to be the same as ammonium oxidation in the above calculation). In particular, the oxygen consumption rate is likely higher in the IOT hadal water than the deep Pacific seawater used in the calculation above, because sediment re-suspension introduces more labile sedimentary organic compound (Nunoura et al., 2013; Wenzhöfer et al., 2016) in addition to the background TOC. The same also applies to the maintenance energy used, which has not been

strictly constrained (Hoehler 2004).



## 5. Conclusion

Using full-depth CTD-CMS observation and sampling, the present study revealed heterogeneous distributions of $CH_4$ within the well-mixed hadal water. We interpret that sediment re-suspension, accompanied by the release of sedimentary compounds into the water column, is the most likely source for the $CH_4$ anomaly as well as other geochemical anomalies in the hadal layer reported in past studies and also observed in the present study. From an energetics viewpoint, microbial ammonium oxidation was shown to only occupy a minor part for the overall microbial oxygen consumption expected in the hadal water column. A great advantage with using the CTD-CMS system to study hadal waters, compared to unconventional methods used in previous studies, is the amount of water that can be sampled using a Carousel sampler. The biogeochemical analyses undertaken in this study only used a ~2L subsample out of 12L seawater sample collected by each Niskin bottle, and the remaining are available for further analyses such as microbiological characterization. To shed the fullest light on the dark hadal biosphere, comprehensive analyses combining hydrography, (bio)geochemistry, and microbiology is indispensable, and full-depth CTD-CMS systems (such as the one available on R/V *Kaimei*) put such studies within our reach.

**Data availability**

Dataset reported will be available on a public website.

**Supplement file**

Supplement Table S1: All analytical results drawn in figures.

Supplement Table S2: Raw data of nitrigication activity measurement

Supplement Figure S1: Full-depth profiles.

**Author contribution**

SK, TN, HU, and TY designed the study. AM, TK, YM, CY, EO, MW, and HU conducted chemical and hydrographic analyses. SK, AM, and TY made a draft. All authors contributed to sampling and gave final approval for submission and publication.

**Competing interest**

The authors declare that they have no conflict of interest.

**Acknowledgement**



The authors thank Katsuhisa Maeno, Yosaku Maeda, the masters, crews, and scientific parties including MWJ and NME staffs of R/V Kaimei cruises (KM16-02, KM16-08, and KM17-01) and a R/V Shinsei-maru cruise (KS-16-8) for their support. Keiko Tanaka, Shoko Tatamisashi, and Dr. Yuichiro Kumamoto respectively supported the CH$_4$, TOC, and radiocarbon analyses. Dr Chong Chen kindly made English editing. Dr Konomi Suda, Dr Yuya Tada, Dr Michinari Sunamura, Miho Hirai, and Yoko Sasaki helped subsampling on board. Dr Kazuya Kitada analyzed and drew seafloor topography.

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



**Tables and Figures**

Table 1: Summary of stations.

| Cruise | Station | YYYYMMDD | Lat degree | Lat minute | Lon degree | Lon minute | Depth (m) | **Relative location |
|---|---|---|---|---|---|---|---|---|
| KM16-02 | 31N01 | 20160505 | 31 | 18.2 | 141 | 46.7 | 5801 | WW |
| KM16-02 | 31N03 | 20160506 | 31 | 20.0 | 142 | 14.5 | 9382 | A |
| KM16-02 | 31N02 | 20160507 | 31 | 18.3 | 142 | 0.3 | 7093 | W |
| KM16-02 | 31N04 | 20160507 | 31 | 19.1 | 142 | 38.9 | 6928 | E |
| KM16-02 | 31N05 | 20160507 | 31 | 19.8 | 143 | 0.1 | 6162 | EE |
| KS-16-8 | 30N01 | 20160708 | 30 | 0.0 | 142 | 36.3 | 9190 | A |
| KS-16-8 | 32N01 | 20160711 | 32 | 9.0 | 142 | 0.0 | 7931 | W |
| KS-16-8 | 32N02 (KEO) | 20160706 | 32 | 21.6 | 144 | 24.2 | 5737 | EE |
| KM16-08 | 34N03 | 20160913 | 34 | 57.6 | 142 | 33.5 | 5837 | E |
| KM16-08 | 34N02 | 20160914 | 34 | 0.1 | 141 | 57.2 | 9224 | A |
| KM17-01 | 29N02 | 20170106 | 29 | 17.6 | 143 | 30.8 | 6062 | E |
| KM17-01 | 29N01 | 20170107 | 20 | 8.9 | 142 | 48.9 | 9820 | A |
| KM17-01 | 26N01 | 20170108 | 26 | 10.3 | 143 | 15.2 | 3306 | (A) |
| KM17-01 | 27N01 | 20170108 | 27 | 30.1 | 143 | 19.1 | 8949 | A |
| KM17-01 | 24N01 | 20170109 | 24 | 16.4 | 143 | 38.3 | 8798 | A |
| KM17-01 | 24N02 | 20170109 | 24 | 15.5 | 144 | 15.0 | 4898 | E |
| | | | | | | | | |
| KH-84-3 | AN1* | 19840823-30 | 29 | 5.0 | 142 | 51.0 | 9768 | A |
| KH-94-3 | LM2* | 19941004-06 | 29 | 4.8 | 142 | 50.0 | 9738 | A |
| | | | | | | | | |
| *Gamo and Shitashima 2018 | | | | | | | | |
| **Relative locations from trench axis. WW: far west, W: west, A: Axis, E: east, EE: far east | | | | | | | | |





Table 2: Summary of analyses.

| | | DO | Nitrate | $\delta^{15}N_{NO3}$ | $\delta^{18}O_{NO3}$ | PO$_4$ | SiO$_2$ | N$_2$O | $\delta^{15}N_{N2O}$ | $\delta^{18}O_{N2O}$ | TOC | $\Delta^{14}$C |
|---|---|---|---|---|---|---|---|---|---|---|---|---|
| | | umol/kg | umol/kg | permil | permil | umol/kg | umol/kg | nM | permil | permil | uM | permil |
| Izu-Ogasawara (this study) | AVG | 164.0 | 35.2 | 5.10 | 2.23 | 2.46 | 145.6 | 17.7 | 8.82 | 51.56 | 37.9 | -206.6 |
| | STD | 0.27 | 0.41 | 0.15 | 0.11 | 0.06 | 0.2 | 1.3 | 0.22 | 0.26 | 1.1 | 4.4 |
| | n | 47 | 47 | 25 | 25 | 23* | 11* | 47 | 47 | 47 | 45 | 40 |
| | | | | | | | | | | | | |
| Izu-Ogasawara (G&S2018) | AVG | 165.7 | 35.1 | NA | NA | 2.39 | ~142 | NA | NA | NA | NA | NA |
| | STD | 0.7 | 0.3 | | | 0.04 | | | | | | |
| | | | | | | | | | | | | |
| Mariana (this study) | AVG | 171.7 | 34.7 | 5.13 | 2.24 | NA | NA | 17.1 | 8.70 | 51.11 | 37.6 | -197.0 |
| | STD | 0.04 | 0.05 | 0.08 | 0.05 | | | 0.8 | 0.10 | 0.24 | 0.6 | 3.9 |
| | n | 10 | 10 | 7 | 7 | | | 10 | 10 | 10 | 10 | 10 |
| | | | | | | | | | | | | |
| | *Data available only from KM16-08 cruise | | | | | | | | | | | |

Figure 1: Seafloor topography of (a) western North Pacific and (b-g) cross section of inner trench. Horizontal red lines in panel (a) represent the sections shown in panels (b-g). Vertical dash lines in panels (b-g) represent CTD-CMS stations (see Table 1).

Figure 2: Vertical profiles of measured parameters. Symbol shapes and colours are classified by locations relative to the axis and latitude, respectively (see Table 1). Horizontal dash lines indicate the abyssal-hadal boundary. Bold horizontal lines shown at the bottom of each panel represent one-sigma analytical errors whereas not shown when the errors are smaller than symbol size.

Figure 3: Vertical profiles of (a) $\delta^{15}N_{NO3}$ increase of cultivated batches from control batch after $^{15}$N-spiked incubation and (b) nitrification activity calculated based on the $\delta^{15}N_{NO3}$ differences. Since the $\delta^{15}N_{NO3}$ increase three times greater than analytical uncertainty (>0.6‰) is regarded as significant, symbols fallen into grey shadow in panel (a) represent insignificant activity. Symbol colours of red, blue, and green respectively represent 15N-labeled substrates of ammonium, urea, and glutamine. Horizontal axes are presented in logarithmic scale.

Figure 4: A DO-$\Delta^{14}$C plot for deep seawater (>4,000 dbar). Symbols with colour are same with those in Figure 2. Filled black circles and a diagonal line respectively represent a dataset from WOCE P01-P03 stations and its fitting line.



Figure 5: Spatial distribution of $\delta^{13}C_{CH4}$ values. Symbols are classified by colour according to relative locations from trench axis. Long horizontal bars indicate the boundary between Izu-Ogasawara and Mariana trenches, Ogasawara Plateau.

Figure 6: A plot between $CH_4$ concentration and $\delta^{13}C_{CH4}$. Symbols with colour are same with those in Figure 2. Open and filled stars respectively represent endmember values assumed for background seawater and inner-trench $CH_4$ source. Broken curves represent theoretical isotope behaviours from the local $CH_4$ source when kinetic isotope effects on microbial methanotrophy ($\varepsilon$) are 5 and 20. A broken-dot curve represents a bimodal mixing between two endmembers.





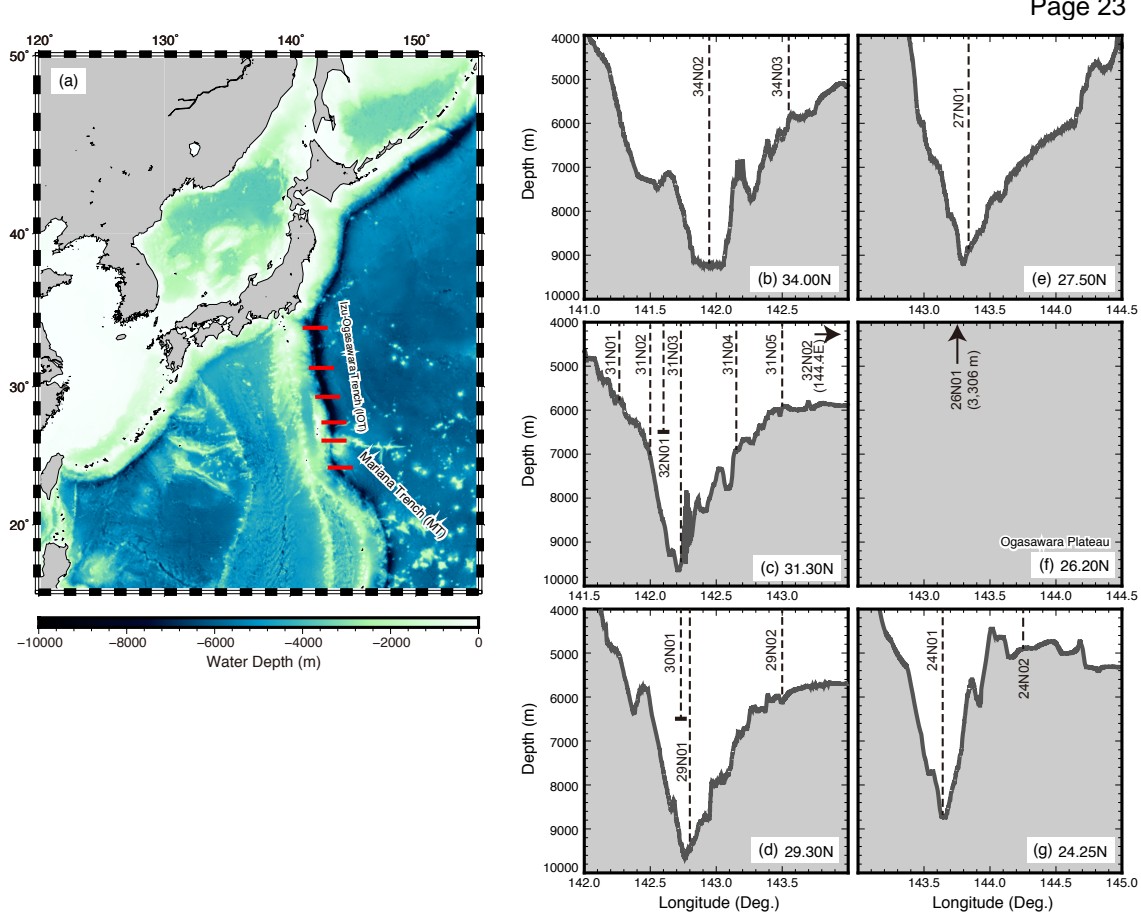

Figure 1 (Kawagucci et al.)





Figure 2 (Kawagucci et al.)





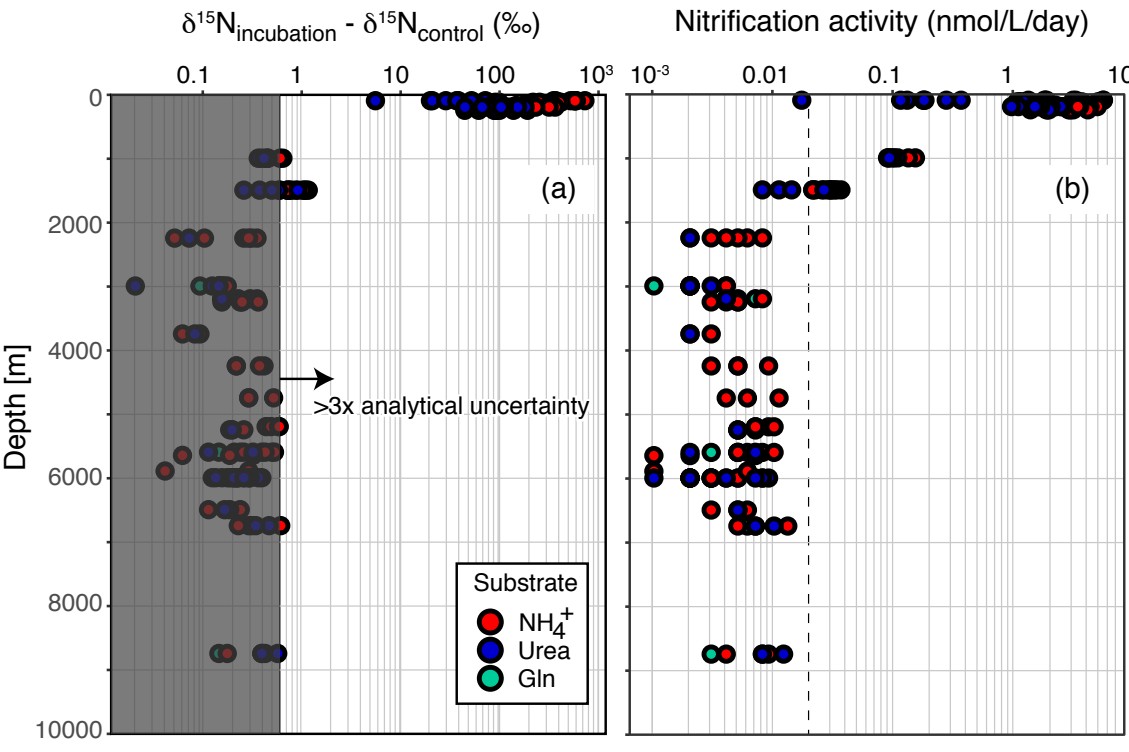

Figure 3 (Kawagucci et al.)



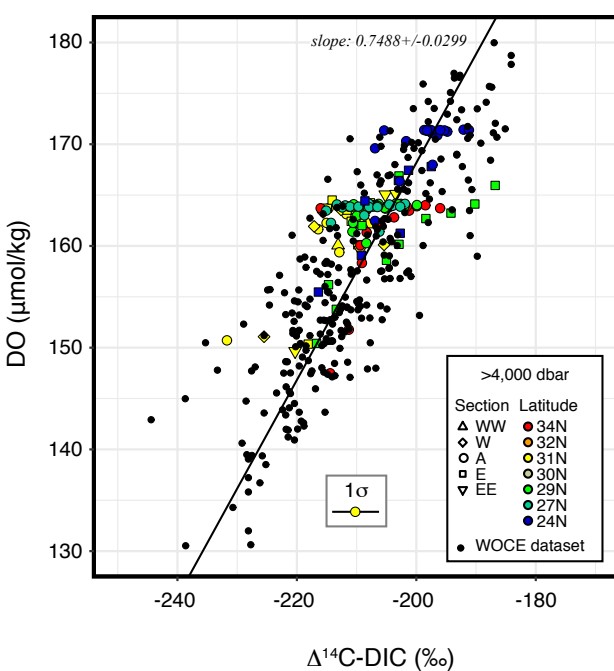

Figure 4 (Kawagucci et al.)

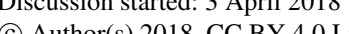



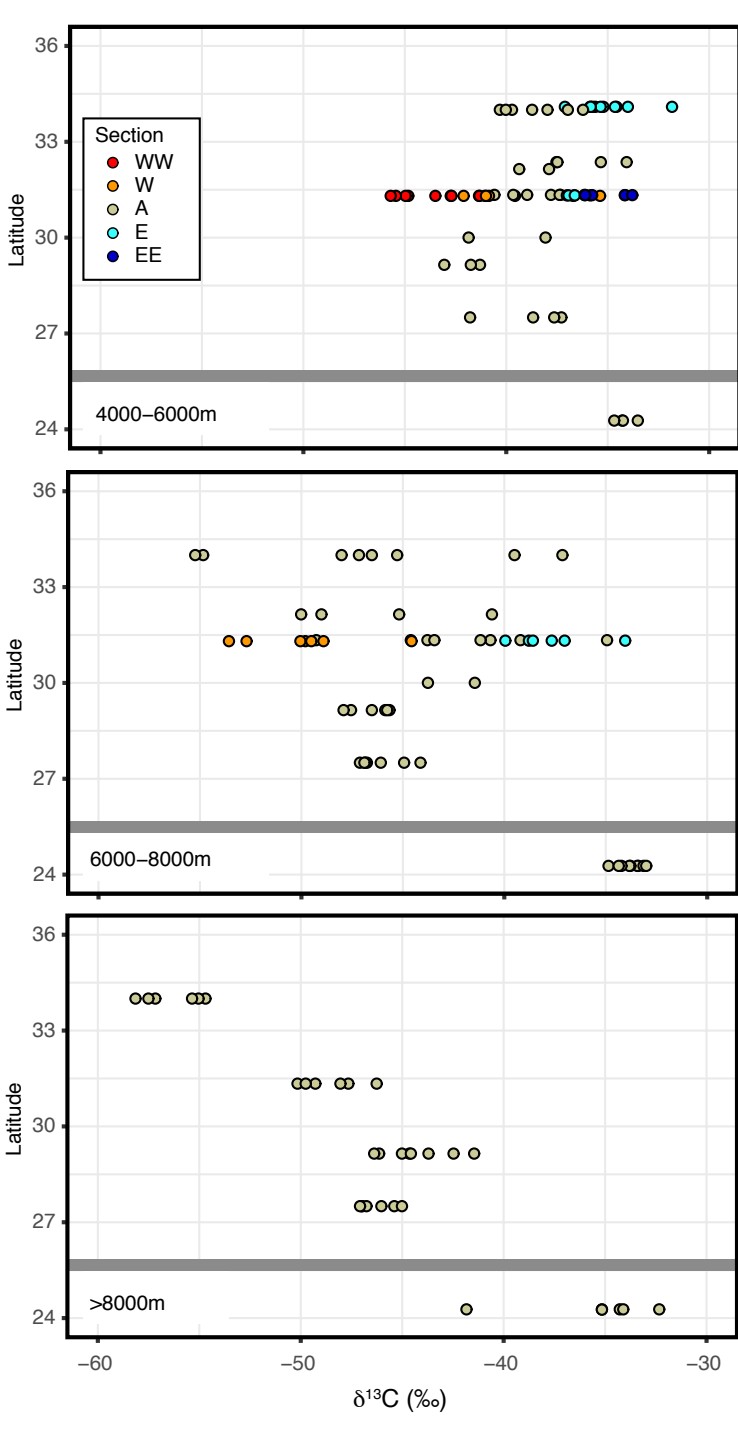

Figure 5 (Kawagucci et al.)





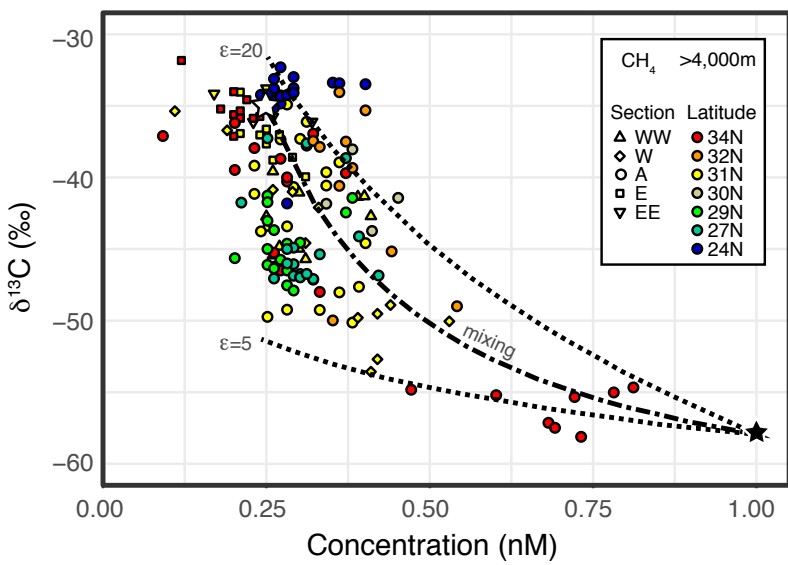

-

Figure 6 (Kawagucci et al.)