# Peer review of "Hadal water biogeochemistry over the Izu-Ogasawara Trench observed with a full-depth CTD-CMS"

_Ocean Science, 2018_

## Referee Comment (RC1) · Anonymous Referee #1 · 30 Apr 2018

This manuscript reports reasonably interesting and unique dataset regarding the hadal water biogeochemistry in the Izu-Ogasawara Trench. Indeed, this kind of dataset is relatively rare I think it should eventually be published, but before that some of my concerns should be appropriately addressed:

The authors state that 1 nM of methane is the threshold for microbial aerobic methanotrophy in Page 9 Lines 17-18, but the measured CH4 concentrations are all <1 nM within hadal depth and they attributed the change in del13 of CH4 to the kinetic isotope fractionation induced by microbial methane consumption in Page 10 Lines 5-10. This is somewhat self-contradictory to me.

[Figure]

I am not fully convinced by the conclusion that the slightly elevated CH4 is due to sediment resuspension because 1). Not all transects within hadal depth show increased CH4 concentrations. Since sediment remobilization is common in hadal sediments, I would expect most of the hadal water has higher than background CH4 concentrations if not all. 2). O2 penetration depths were shown to be around 15-20 cm in hadal sediments (Glud et al., 2013; Wenzhoefer et al., 2016). How thick is the sediment layer do you expect to be removed in order to expose methane to the bottom water? 3). If CH4 is sourced from sediments and oxidized to DIC in the hadal water, I would anticipate to see $\Delta$14C of DIC exhibiting a more depleting trend. But from your profiles, I don't see any old DIC input into bottom water.

The authors provided profiles of N and O isotopes of NO3 and N2O which are indeed quite rare dataset in the deep ocean. But it is a pity the authors did not discuss those data in detail and did not state why they measured them.

Minor comments Page 2 Line 12: People usually take water depth>6000 or 6500 m as hadal (Jamieson et al., 2010; Watling et al., 2013). Better to be consistent with previous publications.

Page 2 Line 29: Rewrite as "Manganese and iron were slightly enriched in the hadal water only and were background levels in the overlying..."

Page 3 Line 15: Rewrite as "Ultimately, a full-depth rated CTD-CMS..."

Page 3 Lines 15-16: This sentence is unclear. Modify it.

Page 4 Line 28: Precision of NH4 analysis is not mentioned.

Page 5 Lines 26-27: This sentence is incorrect. Rewrite it.

Page 6 Section 3.1: There is no big change in potential density anomaly between 6000-7000 m. Instead, a marked gradient change is observed between 6000 m and above. A re-defined hadal water depth is not warranted and is unnecessary to me.

Page 6 Line 24: Rewrite as "which was called the hadal mixed layer". Actually I don't feel comfortable to call it "mixed layer". Because surface mixed layer is a well-mixed layer driven by wind, but what the mechanism for the homogeneous hadal water?

Page 7 Lines 7-8: This sentence is incorrect. Revise it.

Page 9 Line 24: "became constant below 8000 m"? below or above?

Page 11 Lines 25-26: Some more recent studies should be cited, e.g., Luo et al., 2018 Luo, M., Glud, N.R., Pan, B., Wenzhöfer, F., Xu, Y., Lin, G., Chen, D., 2018. Benthic carbon mineralization in hadal trenches: Insights from in-situ determination of benthic oxygen consumption. Geophysical Research Letters, 45. https://doi.org/10.1002/2017GL076232.

Page 12 Line 5: distinct from the abyssal counterpart

Page 12 Lines 24-26: This sentence is incomplete. Rewrite it.

Page 13 Line 5: in previous studies

––––––––––––––––––––––––––––––––

---

## Referee Comment (RC2) · Anonymous Referee #2 · 15 May 2018

The authors present data from the abyssal and hadal Izu-Ogasawara Trench and Marianna Trench using a new CTD-CMS developed for these depths. The data presented are thus novel as such detailed sampling was possible for the first time and their publication is of large interest to the oceanographic and biogeochemical community. In this paper the authors present dual isotope measurements of nitrate and nitrite, nitrous oxide, nitrification rates, TOC, CH4 concentrations and delta13C of methane in addition to oxygen and nutrient concentrations. The paper concentrates on the data from the two trenches but also shows complete water column profiles.

The interpretation of the data set is, however, very sketchy and touches only methane

and its possible sources. For publication the discussion needs to be expanded. The isotopic data of nitrate, nitrite and nitrous oxide require some discussion regarding their evidence for the nitrogen cycle. What do these data reveal about N or O sources and transformation processes? Are these data typical for the region? The methane data are discussed and some ideas about methane sources (and the evidence from C isotopes) are presented. Here my suggestion is to present vertical profiles of these variables at least along 31.30N and 29.3 N in order to substantiate the discussion. Specific comments:

Page 4 line 10: rive reference for the modified Winkler method

Results: are there results of dual isotopes of nitrate only (vs. nitrate+nitrite) can these data provide information about processes of the N cycle?

Page 10 line 3ff: can you name the end member values of delta13C of methane?

Can you specify the transformation processes and isotopic effects on methane delta13C ?

Page 10. Line 16: can the currents be better discerned from vertical profiles/cross sections through the trenches?

Page 11 line 11: does this mean that methane is released from pore waters as sediments are resuspended from the bottom or side of the trenches? Or do these processes take place in the water column after resuspension?

---

## Author Comment (AC1) · 18 May 2018

We really appreciate fruitful comments from anonymous referee #1 on our manuscript. Our reply to referee #1 is below.

We agree with referee#1 on confusing description about a minimum threshold for microbial methanotrophy in Discussion paper. To clarify meaning, we revise it as "CH4 concentrations as low as 0.25 nM are ubiquitous in deep-sea water (Hirota et al., 2010; Son et al., 2014) and are probably regulated by a minimum threshold in microbial uptake for aerobic methanotrophy. " in a revised version of manuscript.

[Figure]

Referee #1 presented concerns about origin of anomalous methane in the hadal watar mass. At first, during this Discussion term, Mn concentrations of the IOT waters were obtained (new Fig 2). The Mn concentrations were more enriched in IOT hadal water than IOT abyssal water, which strongly supports our conclusion in Discussion paper, occurrence of sediment resuspension in the IOT hadal water. We thus add Mn concentrations and their interpretation into Results and Discussion sections of a revised version of manuscript. As stated in Page10Lines3-15 of Discussion paper, variation of CH4 concentrations and isotopic composition in the IOT hadal water were consistently explained by three processes: local CH4 input (from sediment resuspension), microbial CH4 consumption in water column, mixing with background CH4. Even if the resuspension providing CH4 is common in hadal water column, CH4 concentrations could decrease by microbial consumption and the dilution down to ∼0.25 nM (it is shown in Figure 6). Although referee #1 mentioned about thickness of trench bottom sediment, sediment resuspension we expect occurs not only from the axis bottom but also trench slopes (Page11Lines23-24). Even in macro-scale sediment having significant amount of O2, micro-environment within the organic particle could be highly reducing and a situation for local methanogenesis as discussed previously (e.g., Sasakwa etal., 2008). The methanogenesis occurring in such microenvironment within the sediment and its release associated with resuspension would be a possible mechanism by which the observed CH4 anomaly was formed. The D14C-DIC change associated with DIC production through methanotrophy was considered negligible because concentrations of CH4 was at sub-nanomolar level, 6-7 orders of magnitude lower than DIC (2.3 millimolar).

As stated by referee #1, the Discussion parer lucks both description about our motivation for measuring nitrogen molecules and discussion about nitrogen dynamics. Our motivation analyzing nitrogen molecules came from previous microbiological study (Nunoura et al. 2015). That study revealed "hadal biosphere" and considered that the lateral supply of sedimentary organic matter from the trench slope and the relevant nitrogen metabolisms were the key mechanisms to develop the biosphere. We describe

this motivation into the revised version. On the other hand, we cannot further discuss nitrogen dynamics in the hadal water because concentrations and isotope composition were constant demonstrating little processes driving nitrogen cycle there.

About definition of "hadal" depth, this study uses 7,000m based on accurate and precise observation of hydrographic properties with CTD-CMS (Page6Lines23-27). We disagree with referee#1 who stated "no big change in potential density". As stated at Page6Lines23-27, there was significant changes in the density. We believe that the redefinition for the IOT hadal water boundary of 7,000m in this study is reasonable. On the other hand, of course we know that 6,000m or 6,500m are usually used and accepted in general. To avoid confusing readers, we thus add statement into Introduction section as "the hadal waters (deeper than 7,000 m in this study but usually 6,000 or 6,500 m (Jamieson et al., 2010; Watling et al., 2013), see Section 3.1)".

A term "mixed layer" usually means hydrographically homogeneous water mass or like that regardless to the driving force for mixing. In fact, "bottom mixed layer" for example has been used in numerous papers (we can easily find them on Google Scholar). We thus decide to keep using a term "hadal mixed layer" in this study.

We agree with the other minor specific comments, and the manuscript has been revised accordingly.
* * *
[Figure]

[Figure]

Figure 2 (Kawagucci et al.)

**Fig. 1.** new Fig 2

[Figure]

---

## Author Comment (AC2) · 18 May 2018

We really appreciate comments from anonymous referee #2 on our manuscript. Comments are fruitful for us. Our reply is below.

Referee #2 suggests that the manuscript expands interpretation of nitrogen molecules concentrations and isotopic composition. Several sentences presenting preliminary interpretation of the dataset obtained have been added into Results section of the revised version. To be honest, however, it is difficult to expand discussion about nitrogen dynamics due to two aspects. Firstly, for abyssal-hadal waters, concentrations and isotopic composition of N2O and nitrate of the IOT water varied little and were

[Figure]

consistent with those of the deep-sea Pacific water reported so far (Page7Lines16-18 and Lines25-26). It means negligible transformation of nitrogen molecules in the IOT deep-sea water. It is supported by nitrification rates examined that were too low (<0.02 nM/d) to impact on vertical profiles of natively abundant nitrate ($\sim$35uM) and N2O ($\sim$20nM) and their isotope composition. Secondly, for shallow waters (0 - 2,000 m depth), vertical intervals of seawater samplings are too rough (see Figure S1 and Table S1) to provide fruitful discussion about the nitrogen dynamics that are very complicated spatio-temporaly (e.g., Casciotti 2016). Nevertheless, all the analytical results including shallow water are available as TableS1. We believe that the dataset can be contributed for future studies focusing on nitrogen dynamics in shallow water at IOT region or Pacific Ocean if we provide little discussion in this study. Again, a bit more interpretation of nitrogen molecules has been added into Results section of the revised manuscript.

Followings are replies to specific comments.

*A reference and details of DO measurement have been described in the revised version manuscript.

*About name of the d13C value of background CH4, we cannot catch the meaning of referee#2 comment. We name "deep Pacific seawater-sourced background CH4" in the revised manuscript. Transformation processes and isotopic effects on methane d13C in the IOT hadal water have already been discussed at Page3Lines3-15 and Figure 6 while those in the deep Pacific seawater cannot be discussed in this study (it requires basin-scale observation and detection of spatial changes of conc./da13C of CH4).

*Preliminary analyses of hydrographic properties and interpretation from a viewpoint of physical oceanography such as current directions are supportive for our conclusion as well as previous observations. However, detailed analyses are now conducted and will be presented as a separated paper focusing on hydrography (Uchida et al. in

preparation) as stated Page7Lines1-2.

*We are very interested in (but cannot distinguish) whether the hadal water CH4 had been present in porewater as dissolved gas or sediment surface as sorbed gas before the release to water column. For this issue, we will collect sediment cores at trench slopes and axis bottom and analyze CH4 concentrations and isotope composition with different ways for sediment processing.

---

## Author Comment (AC3) · 1 Jun 2018

We are very sorry to miss a Referee #2 comment about interpretations of N and O isotopic composition of nitrate and nitrate+nitrite. We were aware that the original manuscript describe insufficient information about the isotope analyses. For all the samples collected for "nitrate" isotope compositions through the depths, we used the denitrification method without nitrite removal processing and thus obtained d15N(NO3+NO2) and d18O(NO3+NO2) values. In addition to this, we conducted additional isotope analyses for surface seawater in which nitrite was detected significantly at the AA analysis. For these samples, acid injection to removal of nitrite was applied

to obtain d15N(NO3) and d18O(NO3) values. For the selected surface samples, we thus obtained the isotopic compositions of both nitrate+nitrite and nitrate(only). All of our analyses for d15N(NO3+NO2), d18O(NO3+NO2), d15N(NO3), and d18O(NO3) is available at a revised Table S1. As discussed previously in literatures (e.g., Casciotti and McIlvin, 2007; Kemeny et al., 2016), the d15N(NO3+NO2) and d18O(NO3+NO2) values of nitrite-rich samples should be interpreted with caution due to the analytical artifacts. Although Referee #2 provides a suggestion to interpret these isotopic compositions for understandings of N cycle (as a specific comment), it is difficult to expand discussion about nitrogen dynamics due to rough sampling intervals for the surface seawater as stated in the first reply. The revised manuscript includes detailed information about our sample processing and caution for the data interpretation.